# Perceived Work Ability during Enforced Working from Home Due to the COVID-19 Pandemic among Finnish Higher Educational Staff

**DOI:** 10.3390/ijerph19106230

**Published:** 2022-05-20

**Authors:** Saila Kyrönlahti, Subas Neupane, Clas-Håkan Nygård, Jodi Oakman, Soile Juutinen, Anne Mäkikangas

**Affiliations:** 1Faculty of Social Sciences, Unit of Health Sciences, Tampere University, 33014 Tampere, Finland; subas.neupane@tuni.fi (S.N.); clas-hakan.nygard@tuni.fi (C.-H.N.); 2Centre for Ergonomics and Human Factors, School of Psychology and Public Health, La Trobe University, Melbourne, VIC 3086, Australia; j.oakman@latrobe.edu.au; 3Work Research Centre, Faculty of Social Sciences, Tampere University, 33014 Tampere, Finland; soile.juutinen@tuni.fi (S.J.); anne.makikangas@tuni.fi (A.M.)

**Keywords:** ergonomics, stress, musculoskeletal pain

## Abstract

**Background:** Due to COVID-19 pandemic, many employees were forced to suddenly shift to working from home (WFH). How this disruption of work affected employees’ work ability is not known. In this study, we investigated the developmental profiles of work ability among Finnish higher education employees in a one-year follow-up during the enforced WFH. Secondly, we investigated demographic, organizational, and ergonomic factors associated with the developmental profiles. **Methods:** A longitudinal web-survey was conducted with four measurement points (April 2020–February 2021). Employees of a Finnish university who answered the questionnaire at baseline and at least at two follow-up surveys (*n* = 678) were included (71% women, 45% teachers/research staff, 44% supporting staff, 11% hired students). Perceived work ability was measured on a scale of 1–5 in all timepoints. Latent class growth curve analysis was used to identify profiles of work ability. Multinomial logistic regression was used to determine the associations of demographic factors, perceived stress, musculoskeletal pain, functionality of home for work, and organizational support with the work ability profiles. **Results:** Six distinct work ability profiles were identified. For most (75%), work ability remained stable during the follow-up. A total of 17% had a favourable trend (very good-stable or increasing) of work ability, and 8% had non-favourable (poor-stable or decreasing). Poor ergonomics at home, low organizational support, high stress, and musculoskeletal pain were associated with non-favourable development of work ability. **Conclusions:** Heterogeneity in development of work ability during forced WFH was found. Several factors were identified through which work ability can be supported.

## 1. Introduction

In response to the outbreak of COVID-19 disease in spring 2020, national policies on social distancing were placed in most countries, including Finland. The social distancing policies mandated an abrupt shift to working from home (WFH), which has had a profound impact mainly on white-collar workers. Although the prevalence of WFH was increasing, for many, it was not routinely undertaken [1]. According to EU Labour Force Survey, in 2019, before the current pandemic, less than 5% of the EU labour force regularly worked from home [2]. In response to the COVID-19 public health measures, mandatory WFH was instigated across much of the world. On average, WFH was reported by 37% in EU after the pandemic and lockdowns started [3]. The figures for some countries, including Finland, have been much higher (50–60%) [3].

The sudden disruption of work meant very little time for formal measures to be instigated by individual workers and their organizations and the subsequent impact on employee well-being. Prior research has identified that the WFH enables flexibility and an improved work–life interface [4,5]. However, recent studies have also shown negative consequences of mandatory WFH to workers’ well-being. The closure of schools and childcare had a negative impact on those with caring responsibilities [6]. Work–life conflict emerged especially for families with young children, impacting women to a greater extent than men [3,6]. Technostress and work strain increased due to an expansion in the use of digital working tools, especially among workers not accustomed to digital technologies and remote working [7]. Furthermore, for many, the home offices were inadequately equipped for WFH, with poor physical, cognitive, and organizational ergonomic factors, such as inadequate workspace and equipment for work and social isolation from work community, potentially impacting work ability. For example, consequences of poor physical ergonomics may have impacted work ability through musculoskeletal pain [8].

Despite increasing evidence on the impacts of WFH during the COVID-19 pandemic, the impact on employees’ work ability is yet to be studied. Work ability is a comprehensive indicator that describes a worker’s ability to meet the demands of work given the individual’s resources [9]. The individual resources cover a broad and holistic range of factors from a worker’s health and functional ability to competencies, values, and attitudes towards work. Work demands, on the other hand, cover the actual content and demands of work but also the physical and social work environment, community, organization, and management of the work [9]. The different domains of work ability were significantly affected by the mandatory shift to WFH.

The starting points for individual employees vary greatly, with some employees more accustomed and prepared than others to shift to WFH; therefore, a person-centred estimation strategy that considers the possible heterogeneity in the development of work ability during mandatory WFH is warranted. The aim of this study is to investigate the developmental profiles of work ability among Finnish higher education staff during the mandatory WFH from April 2020 to February 2021. The second aim is to study how demographic factors, stress, musculoskeletal pain, the functionality of home for work, and organizational support predict membership in the different work ability profiles.

## 2. Methods

### 2.1. Study Population

The data used in this study were collected as a part of the Well-being 2020 research project, which aimed to explore working at home and its impact on well-being among the staff of Tampere Universities during the coronavirus crisis. The longitudinal study was conducted with four measurement points: in April 2020 (T1), June 2020 (T2), October 2020 (T3), and February 2021 (T4). Data were collected through a web-based questionnaire created with the LimeSurvey.

The study flowchart is presented in Figure 1. At T1, all members of the university community were invited via email to participate in the survey with one reminder. Of the 6929 university employees, 2661 employees responded (response rate 38%). The baseline respondents who were willing to continue their participation in the study received the first follow-up survey with one reminder at T2, resulting in 909 responses. Invitations to participate in the second follow-up survey (T3), with two reminders, were sent to those who had responded to both earlier surveys and agreed to continue their participation. At T3, 692 employees responded. The third follow-up survey (T4) was sent with two reminders to employees who had participated in all earlier surveys and agreed to continue their participation. At T4, 535 employees responded.

The current study uses data from respondents who answered the questionnaire at baseline and at least at two of the three follow-up surveys (*n* = 678). The mean age of participants was 44.3 years (SD = 11.2) at baseline. Educational attainment was most commonly a Master’s degree (48%). As regards the participants’ primary position, 45% were teaching and research staff, 44% were support staff, 8% were doctoral/licentiate students, and 3% were BSc/MSc students at T1.

### 2.2. Measures

#### 2.2.1. Work Ability

To assess work ability the respondents were asked: *“How would you describe your work ability or your ability to make progress towards your degree in the past two weeks?”*(modified from [10]). The respondents were asked to evaluate their work ability on a 5-point Likert-scale from 1 (very poor) to 5 (very good). The same question with same answer options was asked in each of the follow-up rounds but with the following recall periods: *“during past two months”* (T2), *“during autumn 2020”* (T3), and *“in early 2021”* (T4).

#### 2.2.2. Predictor Variables

Baseline characteristics included demographic factors (age, gender, primary position at university, relationship status, current housing situation, number of under-school-aged and school-aged children), ergonomic and organizational factors (functionality of respondents’ home for work, organizational support), and musculoskeletal pain and stress.

The functionality of respondents’ home for work was assessed at baseline with five items (“I have adequate space at home for remote working; “I have the necessary equipment at home for remote working”; “I can find enough peace at home for working”; “I can maintain a healthy work-life balance when working from home”; “My home internet connection works well enough”). Each item was measured on a scale from 1 (strongly disagree) to 5 (strongly agree). Using principal component analyses (PCA), the original items were reduced to one factor (Appendix A), for which the standardized factor score values ranged from −3.7 to 1.4. The factor score was used as a continuous variable in the analyses.

Organizational support was assessed with six items developed for the current study. Four related to university management and support (*“The top management of the university have communicated clearly about the current exceptional circumstances”; “My practical questions have been answered quickly enough”; “I have received enough instructions for performing my tasks and duties from home”; “I have received support for my work when I have encountered difficulties”*). Two items related to the operation of information systems and teleworking tools (*“I have received enough instructions for using the electronic systems and tools such as Teams, Zoom, Panopto, Moodle”; “The electronic systems and tools have worked well technically”*). Respondents indicated their agreement with the statements from 1 (strongly disagree) to 5 (strongly agree). The original items were reduced to one factor (Appendix A), for which the standardized factor score values ranged from −4.4 to 1.4. The factor score was used as a continuous variable in the analyses. Details on the PCA to create the composite variables is described in the Appendix A.

Musculoskeletal pain was assessed at baseline by the question: *“Have you experienced pains, aches or other discomfort in your back, neck, or arms during the past two weeks?”* (modified from [11]). The answer options ranged from 1 (never) to 5 (always) and were recategorized into three classes: low (comprising answer options “never” and “rarely”), moderate (“sometimes”), and high (“often” and “always”). Stress was assessed at baseline by the question *“Stress means you feel tense, restless, nervous of anxious or are unable to sleep because your mind is troubled. Have you been feeling stressed in the last two weeks?”* [12]. The response scale ranged from 1 (not at all) to 5 (very much) and recategorized into low, moderate, and high.

## 3. Statistical Analyses

### 3.1. Trajectory Analyses

Latent class growth curve analysis (LCGA) was used to examine heterogeneity in the development of work ability during the follow-up and to classify individuals into distinct profiles based on their response patterns to the questions about work ability at four timepoints. Work ability at each timepoint was treated as ordinal variables ranging from 1 to 5, with equally spaced levels. Assuming homogeneity of variance within the profiles, the posterior probabilities of belonging to each profile were obtained for each respondent, and they were allocated to the profile for which the probability was the highest [13,14]. The best fit model was chosen based on the interpretation of the identified profiles as well as several statistical model fit criteria (Appendix A) [15]. Models with one to seven classes with a linear and quadratic shape trajectories were examined.

A six-profile solution best fitted the data based on the model fit indices (Appendix A). The six-class solution excluding the quadratic terms was supported by the LMR likelihood ratio test (*p* = 0.030); it ranked best in terms of the highest entropy value (0.79) and the lowest sample-size-adjusted BIC. Although some of the profiles were rather small, they were important in terms of the content. The minimum class size was above 1%, which can be considered adequate [16]. The average posterior probabilities were likewise reasonably high (>0.70) for all profiles. The models were rerun with different starting values to ensure the optimal solution was found. LCGA were run with Mplus software V.7.2 (Mplus, Los Angeles, CA, US). The class assignment information was exported to SPSS v. 26 (IBM), which was used for the explanatory analyses.

Baseline characteristics by the derived work ability profiles are reported as mean and standard deviation (SD) for continuous variables and proportions for categorical variables. Differences between profiles were examined with chi-square test for categorical variables and analysis of variance for continuous variables.

### 3.2. Multinomial Regression Modelling

We used multinomial logistic regression to determine the associations between baseline demographic and ergonomic factors, organizational support, stress, and musculoskeletal pain with the work ability profiles. Odds ratios (OR) with 95% confidence intervals (CIs) were determined for each model. First, each predictor variable was individually examined in univariate regression models using profile membership as a categorical dependent variable. Then, a forward stepwise multinomial regression was used to test which factors significantly (α = 0.10) predicted participants’ work ability when all other variables were mutually adjusted. The variables that survived the selection were simultaneously added into the final model. Multicollinearity was checked using variance inflation factor. Model fit was estimated from Pearson’s goodness-of-fit test. The proportion of variance explained was determined from Nagelkerke’s pseudo R^2^.

Those who did not give information on gender (*n* = 21) were excluded from the explanatory analysis. Furthermore, due to too-small class size (<1%), those who reported their gender as other (*n* = 7) were excluded.

## 4. Results

### 4.1. Work Ability Trajectories

Figure 2 depicts the profiles of work ability. The majority of respondents (52%) belonged to “good-stable” profile, in which work ability remained at a good level across the follow-up. Approximately one-fourth (23%) of respondents were categorized into the “moderate-stable” profile, characterized by stable, moderate level of work ability. “Very good-stable” profile (13% of respondents) reported initially very good work ability, which slightly decreased after T2; yet, the change was not statistically significant (*p* = 0.06 for slope). Our analysis also revealed two rather small work ability profiles in which the slope of change in work ability during follow-up was statistically significant (*p* < 0.05). These small profiles were named “very good-decreasing” (2% of participants), in which participants initially reported very good work ability that decreased to a poor level during follow-up, and “good-increasing” (4% of participants), in which participants reported good work ability at T1, which improved to very good level during follow-up. Finally, a “poor-stable” profile emerged (6% of respondents), in which those reporting poor work ability remained at a poor level across the follow-up.

Because some of the derived work ability profiles were too small to yield reliable results in regression models, we combined some of them into bigger classes. The two profiles showing the most optimal development of work ability (“very good-stable” and “good-increasing”) were combined. Similarly, the profiles showing the least optimal (“poor-stable” and “decreasing”) development of work ability were merged into one class. The “good-stable” profile was chosen as the reference category because it was the most common class.

### 4.2. Baseline Characteristics

The derived profiles of work ability differed in almost all studied factors (Table 1). Those in the less optimal work ability profiles were younger and more often men and teaching/research staff than those in “good-stable” and “very good-stable and good-increasing” profiles. A clear gradient was observed in the two factor-variables (functionality of home for work and organizational support) such that those in the most optimal scored higher than those in the less favourable profiles. Stress and musculoskeletal pain were more prevalent in the less optimal work ability profiles.

### 4.3. Regression Analyses

Table 2 shows the univariate associations of the predictor variables with a membership of the “very good-stable and good-increasing”, “moderate-stable”, and “poor-stable and decreasing” work ability profiles, with the “good-stable” profile as a reference. Age was conversely associated with “moderate-stable” and “poor-stable and decreasing” profiles, while male gender was associated with “poor-stable and decreasing” profile. Having children under school-age was associated with “moderate-stable” profile. Support staff were less likely to belong to “moderate-stable” and “poor-stable and decreasing” profiles as compared to other staff groups. Further, those living in flat or in terraced/semi-detached house were more likely to belong to “moderate-stable” profile than those living in a single-family detached house.

The score for the functionality of home for work was positively associated with membership in “very good-stable and good increasing” profiles and conversely associated with membership in “moderate-stable” and “poor-stable and decreasing” profiles. Organizational support was similarly associated with work ability profiles: the higher the score, the more likely a respondent was to belong to the more optimal work ability profiles. The odds of belonging to “poor-stable and decreasing” or “moderate-stable” profiles were higher for those who reported high work-related stress. High musculoskeletal pain predicted a membership in “moderate-stable” profile, while those reporting high musculoskeletal pain were unlikely to belong to the most optimal work ability profiles.

In the multivariate model (Table 3), increasing age increased the probability of belonging to “moderate-stable” profile. Men were more likely than women to belong to the poorer work ability profiles.

All ergonomic and organizational factors significantly predicted membership of the work ability profiles. One SD increase on functionality of home for work score was associated with OR of 2.60 (95% CI 1.80–3.75) of belonging to “very good-stable and good-increasing” profiles, whereas it decreased the probability of belonging to (the least optimal profiles) “moderate-stable” (OR 0.80; 95% CI 0.63–1.00) and “poor-stable and decreasing” profiles (OR 0.70; 95% CI 0.50–0.97). Similarly, one SD increase on organizational support score was associated with higher probability of belonging to “very good-stable and good-increasing” profiles (OR 1.46; 95% CI 1.09–1.97), whereas it decreased the probability of belonging to “moderate-stable” (OR 0.69; 95% CI 0.55–0.87) and “poor-stable and decreasing” profiles (OR 0.50; 95% CI 0.35–0.70).

High stress predicted membership in “moderate-stable” profile and in “poor-stable and decreasing” profiles. Moderate stress level decreased the odds of belonging to “very good-stable and good-increasing” profiles. Finally, those reporting high musculoskeletal pain had lower odds of belonging to the optimal profiles. Instead, musculoskeletal pain increased the risk of belonging to the “moderate-stable” profile.

Model fitting information showed that the observed and the estimated values did not differ significantly (*p* = 0.17). Chi-square test showed that the fitted model significantly improved the intercept-only model (*p* < 0.001). Approximately 39% of the variation in profiles was explained by the variables included in the final model.

An attrition analysis was conducted to examine baseline differences between the study population and those who were dropped out (*n* = 1658). The results show that women (71% of the study population and 57% of the attrition group χ^2^(3) = 37.06, *p* < 0.001) and support staff (44% of study and 34% attrition group, χ^2^(3) = 28.38, *p* < 0.001) were overrepresented in the study population. The study population reported a higher mean work ability (3.74 vs. 3.59, *p* < 0.001) than the attrition group. The groups did not differ in age, t(1327.35) = 1.96, *p* = 0.05.

## 5. Discussion

The present longitudinal study used repeated questionnaires to identify developmental pathways of work ability among white-collar workers during exceptional circumstances that required the employees to suddenly shift to WFH. We found that among approximately half of the university staff, work ability remained at a good-stable level throughout the one-year follow-up, whereas near to one-fourth (23%) of the respondents reported a stable moderate level of work ability. Two small profiles with less optimal work ability profiles were also found: one that showed a steep decline of work ability from very good to a poor level and another for whom work ability remained at a stable poor level. Despite the exceptional situation, for 4% of the respondents, work ability improved during the follow-up period.

The longitudinal design and the use of the person-centred method in analysing the development of work ability is a major strength of our study. A traditional variable-centred approach could not have captured the individual variability in the development of work ability. Our results contribute to an understanding of how employees’ work ability is affected by sudden changes in their work demands and resources. The results indicate that job-related well-being experiences during enforced remote work diverge significantly, and for some employees, remote working has been difficult. Recent evidence is in accordance with our results, which demonstrated heterogeneity in the development of work well-being during the enforced remote work among white-collar workers [17,18].

We also identified factors that predicted membership in the different work ability profiles. Of the demographic factors, younger age was associated with the least optimal work ability profiles. This contradicts some of the earlier findings, which, overall, have shown that age is reversely associated with work ability [19]. On the other hand, recent evidence has shown that younger people report lower levels of well-being during the pandemic, with lower overall levels of life satisfaction and optimism and a greater risk of depression as compared to older people [3]. Our results also showed that male gender predicted less optimal work ability profiles. Previous studies suggest that the sociopsychological consequences of the COVID-19 mandated lockdown affected women’s psychological health more strongly than men’s [3,20] owing partly to increasing caring responsibilities during the lockdown, which has given rise to increasing work–life conflicts [3,4,19,21]. In our data, most respondents did not have children, which may explain the difference in results.

It was less surprising that a higher level of stress and musculoskeletal pain predicted poorer work ability. The adverse effects of stress on workers’ health are well-documented (e.g., [22,23]) but according to our knowledge, the effects on work ability in a WFH context have not been studied before. Similarly, previous evidence has shown that musculoskeletal pain is associated with reduced work ability [24,25] and that the prevalence of musculoskeletal pain increased after switching to WFH during the COVID-19 pandemic [26,27]. In line, our results show that the prevention of stress and musculoskeletal pain are key to maintaining and promoting work ability. The suitability of one’s home for WFH is a key resource for safe and productive working. Increased musculoskeletal pain during a COVID-19 pandemic may, in part, signal poor physical working arrangements at home. Further, telecommunication connections and software suitable for teleworking are essential preconditions for effective teleworking. In our study, a high score on the variable encompassing a range of factors important for WFH ergonomics was associated with increased likelihood of belonging to the “very good-increasing” work ability profile. Lower scores, on the other hand, significantly increased the risk for less favourable work ability profiles.

Employees’ experience of the support provided by the organization is a significant work resource that promotes commitment and work performance [28,29]. In line, our results showed that respondents’ experiences of organizational support provided during the forced WFH was associated with work ability. Sufficient support increased the likelihood of belonging to the most optimal work ability profiles, while perceived insufficient support predicted non-optimal work ability profiles.

### Limitations

Selection bias may have affected the results of our trajectory analyses, as the attrition analyses revealed that those who continued participating in the study after baseline survey had better work ability at baseline as compared to those who dropped out. The baseline situation strongly predicted the development of work ability; therefore, the proportion of participants in the least optimal work ability profiles may be underestimated.

Another limitation is that data collection commenced during the COVID-19 lockdown in April 2020, and we did not adjust our analyses for any pre-pandemic factors. In particular, the fact that some of the employees may have been more accustomed and therefore better prepared to WFH than others may have affected the work ability profiles found as well as the observed associations. The majority of the sample (65%), however, did not have previous remote work experience prior to the COVID-19 pandemic, as approximately one-third of the participants had not worked remotely at all, and 40% had worked remotely less than one day per week. [18] Moreover, the COVID-19 pandemic situation itself gave rise to health concerns and mandated social isolation, which undoubtedly affected the respondents’ work ability.

The COVID-19 situation mandated WFH, but it has been previously suggested that a tailored WFH organizational policy, in which employees’ needs and preferences for WFH are considered, is an optimal approach to facilitate employees’ well-being [30]. The results of our exploratory analyses provide insights on the factors that are important in promoting good work ability when working from home. Future studies are warranted to investigate the mechanisms through which the identified predictors of work ability operate.

## 6. Conclusions

To our knowledge, this was the first study to investigate the development of work ability and its predictors among white-collar workers during the WFH mandated by the COVID-19 public health restrictions. For most employees, work ability was maintained across the follow-up, but heterogeneity in the development of work ability indicates that individual starting points for WFH should be considered. Functionality of employees’ home for work with adequate physical, cognitive, and organizational ergonomics are important in maintaining work ability while working from home. The results can advise organizations to optimize multi-location work conditions in the future. Means to provide workers with a functional work environment and adequate organizational support while working from home in order to promote white-collar workers’ work ability should be considered.

## Figures and Tables

**Figure 1 ijerph-19-06230-f001:**
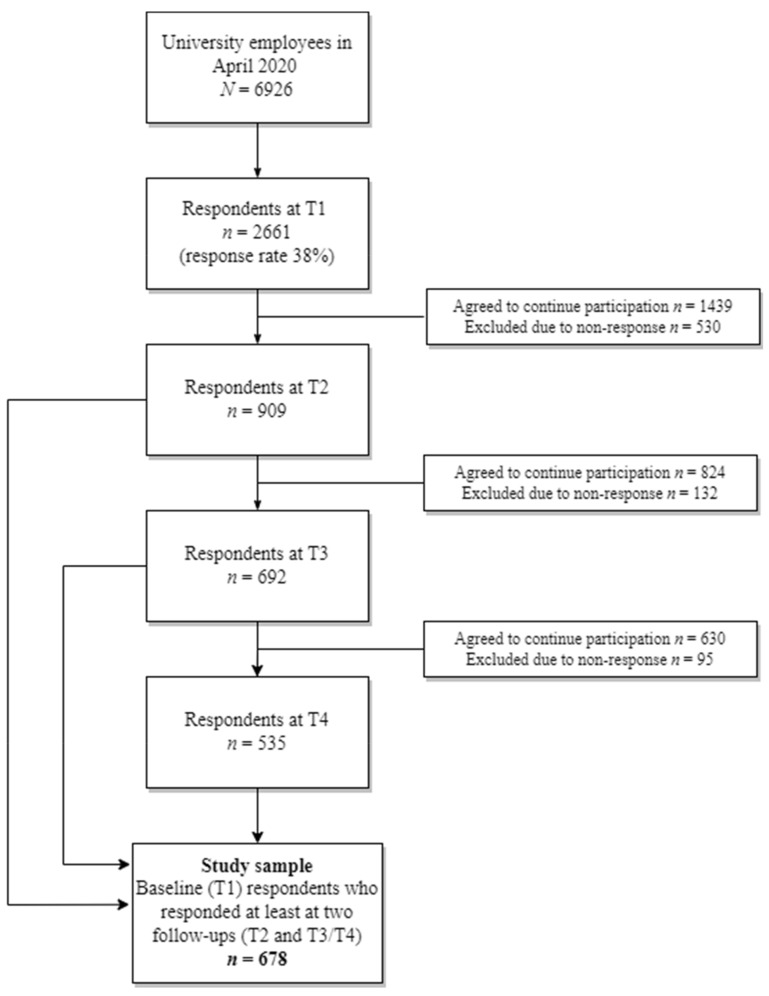
Flowchart of university employees included in the study of work ability profiles during COVID-19 lockdown. T1–T4 are data collection points: T1, April 2020; T2, June 2020; T3, October 2020; T4, March 2021.

**Figure 2 ijerph-19-06230-f002:**
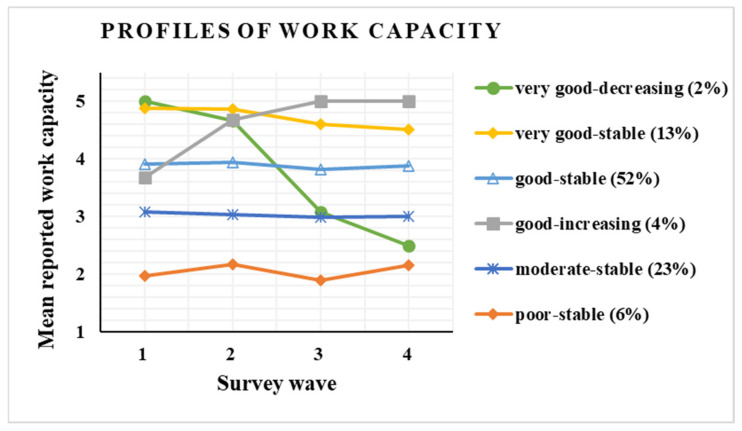
Profiles of work ability during COVID-19 lockdown among university employees (number of participants in each profile: high-decreasing = 12, high = 87, good = 375, good-increasing = 26, moderate = 157, poor = 39). T1–T4 are data collection points: T1, April 2020; T2, June 2020; T3, October 2020; T4, March 2021.

**Table 1 ijerph-19-06230-t001:** Baseline characteristic of the study population and each of the derived work ability profiles among university staff (*n* = 678).

	All (*n* = 678)	Very Good-Stable and Good-Increasing (*n* = 113)	Good-Stable (*n* = 357)	Moderate-Stable (*n* = 157)	Poor-Stable and Decreasing (*n* = 51)	*p* for Difference
**Demographic factors**						
Age, years, mean (SD)	44.3 (11.2)	46.6 (11.0)	45.4 (11.3)	41.2 (10.4)	40.7 (11.3)	<0.001
Gender, %						0.031
Women	75	76	74	65	55	
Men	21	22	22	30	41	
Other/prefer not to say	4	2	4	5	4	
Primary position, %						<0.001
Teaching/research staff	45	37	42	56	53	
Support staff	44	57	49	30	22	
Doctoral/licentiate student	8	5	7	10	22	
BSc/MSc student	3	2	3	5	4	
Relationship status, %						0.056
Single	17	13	15	22	26	
In a relationship	83	87	85	78	74	
School-aged children (yes %)	29	31	28	32	24	0.669
Children under school-age (yes %)	18	12	17	25	14	0.033
Current housing						0.016
Single-family detached house	34	42	36	24	25	
Flat	44	36	42	49	57	
Terraced/semi-detached house	23	22	22	27	18	
**Ergonomic and organizational factors**						
Functionality of home as workplace, mean (SD) ^a^	0.0 (1.0)	0.69 (0.71)	0.06 (0.89)	−0.43 (0.98)	−0.60 (1.28)	<0.001
Organizational support ^b^, mean (SD)	0.0 (1.0)	0.50 (0.83)	0.11 (0.91)	−0.35 (0.91)	−0.82 (1.35)	<0.001
Musculoskeletal pain, %						<0.001
Low	49	71	50	35	39	
Moderate	19	18	19	16	24	
High	32	11	31	49	37	
Work-related stress, %						<0.001
Low	51	81	54	27	31	
Moderate	22	10	25	30	8	
High	27	9	21	43	61	

Note. SD, standard deviation. Summary statistics calculated among participants with non-missing data. Missing values included: age *n* = 5, gender *n* = 2, primary position *n* = 2, relationship status *n* = 22, current housing *n* = 3, under-school-aged children *n* = 5, and school-aged children *n* = 5. ^a^ Standardized factor score, range from −3.7 to 1.4. ^b^ Standardized factor score, range from −4.4 to 1.

**Table 2 ijerph-19-06230-t002:** Univariate associations between profiles of work ability among university staff during COVID-19 lockdown with baseline predictors. Multinomial logistic regression analysis odds ratios (OR) and 95% confidence intervals (CI).

	Very Good-Stable and Good-Increasing vs. Good-Stable	Moderate-Stable vs. Good-Stable	Poor-Stable and Decreasing vs. Good-Stable
	OR (95% CI)	OR (95% CI)	OR (95% CI)
**Demographic factors**			
Age	1.01 (0.99–1.03)	**0.97 (0.95–0.98)**	**0.97 (0.94–0.99)**
Gender			
Women	ref.	ref.	ref.
Men	0.93 (0.55–1.56)	1.52 (0.99–2.32)	**2.47 (1.33–4.58)**
Primary position			
Teaching/research staff	ref.	ref.	ref.
Support staff	1.29 (0.82–2.03)	**0.51 (0.33–0.77)**	**0.37 (0.18–0.78)**
Doctoral/licentiate student	0.73 (0.26–2.04)	1.13 (0.56–2.27)	2.31 (0.99–5.39)
BSc/MSc student	0.64 (0.14–3.00)	1.15 (0.43–3.07)	1.01 (0.21–4.81)
School-aged children			
No	ref.	ref.	ref.
Yes (one or more)	1.05 (0.65–1.68)	1.06 (0.70–1.62)	0.78 (0.39–1.56)
Children under school-age			
No	ref.	ref.	ref.
Yes (one or more)	0.58 (0.30–1.12)	**1.63 (1.03–2.59)**	0.78 (0.34–1.82)
Relationship status			
Single	ref.	ref.	ref.
In relationship	1.22 (0.65–2.30)	0.64 (0.39–1.04)	0.50 (0.25–0.99)
Current housing			
Single-family detached house	ref.	ref.	ref.
Flat	0.74 (0.46–1.21)	**1.96 (1.22–3.16)**	1.82 (0.90–3.67)
Terraced/semi-detached house	0.86 (0.48–1.54)	**2.14 (1.25–3.69)**	1.20 (0.49–2.93)
**Ergonomic and organizational factors**			
Functionality of home as workplace	2.94 (2.11–4.10)	**0.63 (0.51–0.77)**	**0.53 (0.39–0.71)**
Organizational support	**1.76 (1.33–2.32)**	**0.61 (0.45–0.74)**	**0.41 (0.31–0.55)**
Work-related stress			
Low	ref.	ref.	ref.
Moderate	**0.24 (0.12–0.49)**	**2.29 (1.40–3.75)**	0.58 (0.19–1.81)
High	**0.27 (0.13–0.57)**	**4.09 (2.53–6.59)**	**5.45 (2.77–10.75)**
Musculoskeletal pain			
Low	ref.	ref.	ref.
Moderate	0.61 (0.34–1.08)	1.11 (0.63–1.93)	1.44 (0.65–3.18)
High	**0.25 (0.13–0.47)**	**2.10 (1.37–3.22)**	1.60 (0.81–3.15)

Note: ref. indicates the reference group.

**Table 3 ijerph-19-06230-t003:** Multivariate associations between profiles of work ability among university staff during COVID-19 lockdown with baseline predictors. Multinomial logistic regression analysis OR and 95% confidence intervals (CI).

Predictor	Very Good-Stable and Good-Increasing vs. Good-Stable	Moderate-Stable vs. Good-Stable	Poor-Stable and Decreasing vs. Good-Stable
	OR (95% CI)	OR (95% CI)	OR (95% CI)
**Individual/background factors**			
Age	0.98 (0.96–1.00)	**0.97 (0.95–0.99)**	0.98 (0.94–1.02)
Gender			
Women	ref.	ref.	ref.
Men	0.78 (0.42–1.42)	**1.73 (1.05–2.84)**	**2.53 (1.23–5.21)**
Primary position			
Teaching/research staff	ref.	ref.	ref.
Support staff	0.98 (0.58–1.66)	0.60 (0.15–1.96)	0.98 (0.16–6.05)
Doctoral/licentiate student	0.99 (0.58–1.66)	**0.60 (0.37–0.96)**	0.59 (0.26–1.31)
BSc/MSc student	0.35 (0.11–1.14)	0.91 (0.39–2.12)	2.84 (0.96–8.44)
**Ergonomic and organizational factors**			
Functionality of home as workplace	**2.60 (1.80–3.75)**	**0.80 (0.63–1.00)**	**0.70 (0.50–0.97)**
Satisfied with the activities of Tampere University	**1.46 (1.09–1.97)**	**0.69 (0.55–0.87)**	**0.50 (0.35–0.70)**
Work-related stress			
Low	ref.	ref.	ref.
Moderate	**0.28 (0.13–0.60)**	**2.17 (1.26–3.71)**	0.51 (0.15–1.65)
High	0.50 (0.23–1.12)	**2.98 (1.74–5.12)**	**3.57 (1.63–7.79)**
Musculoskeletal pain			
Low	ref.	ref.	ref.
Moderate	0.86 (0.46–1.63)	1.00 (0.55–1.84)	1.59 (0.66–3.88)
High	**0.38 (0.18–0.77)**	**1.82 (1.11–2.98)**	1.35 (0.61–2.99)

Note: Stepwise forward variable selection. α = 0.10. Chi-square *p*-value for model fit < 0.001 (273.847 with 33 degrees of freedom). Nagelkerke value 0.391; ref. indicates the reference group.

## Data Availability

The data are available from A.M. upon reasonable request.

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
