# Peer review of "Perceived Work Ability during Enforced Working from Home Due to the COVID-19 Pandemic among Finnish Higher Educational Staff"

_ijerph, 2022, doi:10.3390/ijerph19106230_

Round 1

Reviewer 1 Report

This is a well written and clear paper. Thus, only some general comments:

  • Regarding the method and statistical analyses, the selected person-centered approach seems to be useful due to the different individual starting points of the outcome variable.
  • The applications of the selected statistical procedures seem to be suitable for testing the hypotheses: thereby, the use of the LCGA (e.g., Berlin et, 2014; Nylund et al., 2007) and the multinominal regression modelling appear appropriate regarding the assumptions for these analyzes and the present data set (e.g., El-Habil, 2012)
  • Furthermore, the choice of measurement points seems to represent the fluctuations of the pandemic during the summer and winter months.
  • Optionally, tables 2 and 3 could also be displayed in landscape format or at least on one page for better clarity.
  • Finally, the assessment of "stress" could have been improved by the usage of an established inventory or some more items. Now: It is as it is, but this limitation should be a bit elaborated on.

El-Habil, A. M. (2012). An application on multinomial logistic regression model. Pakistan Journal of Statistics and Operation Research, 271-291. doi: 10.18187/pjsor.v8i2.234

Berlin, K. S., Parra, G. R., & Williams, N. A. (2014). An introduction to latent variable mixture modeling (part 2): longitudinal latent class growth analysis and growth mixture models. Journal of Pediatric Psychology39(2), 188-203. doi: 10.1093/jpepsy/jst085

Nylund, K. L., Asparouhov, T., & Muthén, B. O. (2007). Deciding on the number of classes in latent class analysis and growth mixture modeling: A Monte Carlo simulation study. Structural equation modeling: A Multidisciplinary Journal14(4), 535-569. doi: 10.1080/10705510701575396

Reviewer 2 Report

An interesting longitudinal study of great importance in the context of COVID and the enforced WFH landscape. It would be interesting to compare this study on university staff/students to other white collar industries to see if there are any differences. 

What was lacking was a discussion about the pre-COVID context for university staff, particularly to what extent their work was conducted in the online landscape before the pandemic. This is likely to have influenced the profiles of work capacity. 
